# Genetic Structure of the Ca Rater Mallorquí Dog Breed Inferred by Microsatellite Markers

**DOI:** 10.3390/ani12202733

**Published:** 2022-10-11

**Authors:** Lourdes Sofía Aguilera García, Amado Manuel Canales Vergara, Pedro Zurita Herrera, José Manuel Alanzor Puente, Águeda Laura Pons Barro, Susana Dunner, Carlos San José Marques, Juan Vicente Delgado Bermejo, Amparo Martínez Martínez

**Affiliations:** 1Department of Genetics, Faculty of Veterinary Sciences, University of Córdoba, 14071 Córdoba, Spain; 2Animal Breeding Consulting S.L., 14014 Córdoba, Spain; 3Institud de Reserca i Formació Agroalimentaria i Pesquera de les Illes Balears (IRFAP), Govern Illes Balears, 07009 Palma, Spain; 4Departamento de Producción Animal, Universidad Complutense de Madrid, 28040 Madrid, Spain; 5Biodonostia Instituto de Investigación Sanitaria, Donostia, 20014 Gipuzkoa, Spain

**Keywords:** molecular markers, genetic diversity, population genetic structure, genetic distance, dog breeds, conservation

## Abstract

**Simple Summary:**

Ca Rater Mallorquí is a hunting and ratting dog that prevents disease spread and economic losses related to rodent activities on farms. Their presence on farms decreases the need for chemical products against rats and other rodents. Unfortunately, the breed’s number of births has declined. We aimed to study the intra and interracial characterisation of the Ca Rater Mallorquí dog breed, clarify its genetic structure and relationship to other related canine breeds, and find preservation solutions for the breed. Our results clearly show that it is a genetically independent and well-defined breed. However, according to census data, this breed should be part of a conservation plan. Our study provides essential knowledge of the Ca Rater Mallorquí genetic situation and constitutes the starting point for designing and implementing a conservation plan to avoid extinction.

**Abstract:**

Ca Rater Mallorquí is a dog breed from the Island of Mallorca (Spain) traditionally used as a hunting and ratting dog to prevent disease spread and economic losses related to rodent activities on farms. However, the census data shows a population decline that should be addressed by implementing a conservation program. The first step to implementing a conservation plan is knowing the genetic situation of the Ca Rater Mallorquí population. Therefore, we aimed to genetically characterise the breed in our study. We analysed 33 microsatellites recommended by the International Society of Animal Genetics (ISAG) in 77 samples. Data were obtained from 13 samples of Balearic, Spanish, and international dog breeds to study the genetic diversity among breeds. The population did not significantly deviate from the Hardy–Weinberg equilibrium with heterozygosity (Ho) of 0.655 and expected heterozygosity (He) of 0.685. The Wright’s fixation indices, the Factorial Correspondence Analysis (FCA), a dendrogram representing Reynolds genetic distance between populations, and the pairwise F_ST_ values establish the Ca Rater Mallorquí as an independent breed distinct from the Balearic, Spanish, and international breeds.

## 1. Introduction

The conservation of animal genetic resources (anGR) has been a major concern for the international community since the 1992 Convention on Biological Diversity, which set out the following aims: the conservation of biological diversity; the sustainable use of its components; and fair and equitable sharing of the benefits arising from its sustainable use of genetic resources [1]. Despite the Global Plan of Action for the Conservation of Animal Genetic Resources, the Interlaken Declaration [2], the Strategic Plan for Biodiversity 2011–2020, or the Aichi Biodiversity Targets consisting of a global partnership to reach "Zero Extinction" [3], it has not always been possible to halt biodiversity losses. These losses are especially acute between local breeds, despite being well-adapted to their surrounding environment, and part of our genetic and cultural heritage. Nonetheless, it does not help that the conservation status of these local breeds remains unknown and without an established conservation programme. The Spanish canine breed Ca Rater Mallorquí is one of these cases, according to the Domestic Animal Diversity Information System (DAD-IS) [4].

This genetic erosion is especially hard on local dog breeds, where the transformation of functional dog breeds to simple pets has led several breeds to near extinction and denaturalization. Today, the new requirements to fight against global change demand the use of no contaminant resources in human practices.

Ca Rater Mallorquí is an excellent example of a breed in human service, particularly as a rat-hunting dog, to avoid the use of rodenticides and other chemical products, thereby preserving the dissemination of microbial and parasitic diseases. They are a native breed from Mallorca Island, historically linked to farming. This breed is strongly related to Gos Rater Valencià due to the arrival of rice growers from Valencia to Mallorca and the repopulation of Valencian villages with Balearic citizens who brought their dogs [5]. Furthermore, the morphological characteristics of this breed have linked it to two ancient dog breeds: dogs with pharaonic origin and dogs with the terrier suffix [6]. 

Its hunting and ratting abilities have often linked the Ca Rater Mallorquí with terrier dogs, despite their relationship not being proven genetically. On farms, they are used as ratting dogs to prevent the spread of diseases and economic losses related to rodents’ activities on farms. Its presence translates into a decline in chemical products applied to farms against rats and other rodents. These economic and public health profits should help prevent the breed’s extinction risk. 

If we consider the census, 146 out of 363 registered animals were founders of the breed. The maximum number of females was 33, and the maximum number of progenies per male was 72. Since 2011, a continued decline in births has been observed [7]. Fortunately, balanced conservation methods have been used to maintain their purity, functionality, and efficiency, positively contributing to the breed’s diversity. In fact, Alanzor et al. [7] have differentiated subpopulations according to the breeders, owners, and locations. However, it remains to be tested if these subpopulations are genetically sound.

Microsatellites are molecular markers that investigate genetic diversity and establish genetic relationships among dog breeds, as demonstrated by several studies with guard dogs, including Italian shepherd dogs [8] from the Western Balkans [9], African village dogs [10], Czech Spotted dogs [11], and Podenco Valenciano [12]. Furthermore, their use in genetic structure studies [13], parentage tests, and dog breed affiliations [14] has been proven. Although SNPs are the current markers of election for these kinds of studies [15,16,17], microsatellites are informative and suitable markers for local breeds, as it is well-known that current commercial SNP chips underestimate diversity and distort relationship estimates [18]. Another important point to consider is that genotyping populations with high-throughput SNP panels is still quite expensive and unaffordable without any external funding, as in this research. Finally, the database used to compare Ca Rater Mallorquí dogs to other Spanish and international breeds was built with microsatellites, as recommended by the FAO [2].

A conservation program should be designed and implemented due to the findings of the census data. The first step is to conduct a genetic characterisation using microsatellites, a tool that has proven accurate and efficient [7,19]. In this study, we used a list of microsatellites created by the ISAG for genetic diversity analysis in the canine species [20]. Our results showed that the Spanish canine breed Ca Rater Mallorquí is a well-defined and genetically homogeneous breed with moderate genetic diversity. A consolidated genetic profile will support the recuperation of unregistered animals who are genetically pure using genetic assignment techniques.

## 2. Materials and Methods

### 2.1. Animal Records 

A total of 77 hair samples from the Ca Rater Mallorquí breed were analysed. All the samples were derived from the island of Mallorca and obtained by specialised veterinarians during their routine practice of official identification and sanitation programmes. The animals phenotypically complied with the breed standard in different morphological competitions or exhibitions organised by the Ca Rater Mallorquí Club. 

To study the genetic diversity among the Ca Rater Mallorquí and Balearic, Spanish, and international breeds, we used data from 17 breed samples (Table 1) from the Sample Bank of the Applied Molecular Genetics Laboratory of the PAIDI-AGR-218 Research Group at the University of Cordoba. The breeds included Ca de Bestiar, Ca de Bou, Ca de Conils de Menorca, Ca Mè Mallorquí, and Ca Eivissenc, or Podenco Ibicenco, from the Balearic Islands. The Terrier breed genotypes were provided by the Animal Nutrigenomics group of the Department of Animal Production at the Complutense University of Madrid, Spain (Table 1).

### 2.2. DNA Extraction

DNA were extracted using Chelex-100^®^ resin (Bio-Rad, Hercules, CA, USA) according to Walsh’s method [21]. Three hair bulbs per sample were used for the extraction. 

### 2.3. Microsatellite Amplification

We used 33 microsatellites recommended by the International Society of Animal Genetics (ISAG) for diversity and parentage studies in dogs (https://www.isag.us/Docs/AppGenCompAnim2019.pdf (accessed on 13 May 2022)). These markers are distributed in a core panel (AHT121, AHT137, AHTh130, AHTh171, AHTh260, AHTk211, AHTk253, CXX279, FH2848, FH2054, INRA021, INU005, INU030, INU055, REN105L03, REN162C04, REN169D01, REN169O18, REN247M23, REN54P11, and REN64E19) and an additional panel (2642RD, 1404RD, 1878RD, 0914RD, 2469RD, 0176RD, 0959RD, 0323RD, 0669RD, 0123RD, 1055RD, and 1257RD). Details about the markers can be observed in Appendix A.

Polymerase chain reaction (PCR) amplified the microsatellites in several multiplex reactions. For a final volume of 10 μL, we added 6 μL of master mix (2 μL of MyTAQ 5X, 0.1 μL of MYTAQ HS, 0.5 μL of 2.7 mM Betaine, and 3.4 μL of distilled water per sample), 2 μL of each primer mix (primers diluted in 10nM Tris at concentrations between 0.2 and 0.7 M) and 2 μL of DNA. Multiplex reactions were amplified with the following programme: 5 min at 95 °C; 35 cycles of 30 s at 95 °C; 90 s at 60 °C; 60 s at 72 °C; the final phase of 20 min at 72 °C. For the P5 multiplex, the PCR cycles were slightly different: 40 s at 95 °C; 60 s at 55 °C; 60 s at 72 °C; the final phase of 30 min at 72 °C.

### 2.4. Polymorphism Detection

Fragments obtained from PCR were separated by electrophoresis using an ABI3130Xl automatic capillary sequencer (Thermofisher Scientific, Madrid, Spain) using Genescan® 500 HD LIZ Orange (Thermofisher Scientific, Madrid, Spain) as the size standard. The fragments and allelic typing analyses were developed using the software Genescan Analysis® v. 3.1.2 (Thermofisher Scientific, Madrid, Spain) and Genotyper® v 2.5.2 (Thermofisher Scientific, Madrid, Spain), respectively. Genotyping was conducted in two laboratories at the Animal Breeding Consulting S.L. (Córdoba, Spain) and Universidad Complutense de Madrid (Madrid, Spain). The standardisation of the genotyping in both laboratories was guaranteed since they regularly participate in the ISAG Dog Comparison Test (https://www.isag.us/comptest.asp (accessed on 28 March 2020)).

### 2.5. Statistical Analysis

The genetic diversity within the dog populations of Ca Rater Mallorquí and among the Ca Rater Mallorquí, Balearic, Spanish, and international breeds, was studied.

We used CERVUS v. 3.1 (Field Genetics, Bozeman, MT, USA) [22] to obtain the number of alleles (NA), expected heterozygosity (He), observed heterozygosity (Ho), and the polymorphic information content (PIC) per marker in the Ca Rater Mallorquí. The Hardy−Weinberg equilibrium was tested using the program GENEPOP v. 3.1c (Michel Raymond and Francois Rousset, Montpellier, France) [23]. Results were obtained for the 33 microsatellites.

In the between-breed study, only the core panel of markers was used, and the following parameters were determined: the mean number of alleles per locus (MNA), expected heterozygosity (He) and observed heterozygosity (Ho) per population were calculated with the MICROSATELLITE TOOLKIT software for Excel (V. 3.1, Park, Dublin, Ireland) [24] and the effective number of alleles with POPGENE v. 1.32 (University of Alberta, Edmonton, AA, Canada) [25]. FSTAT v. 2.9.3.2. (Jérôme Goudet, Lausanne, Switzerland) [26] calculated the average allelic richness (Rt) for each population, using a sample size of eight diploid individuals. 

According to Weir and Cockerham [27], GENETIX v 4.05 (Belkhir K, Montpellier, France) [28] was used to obtain Wright’s fixation indices (F_IS_ or within-breed inbreeding coefficient; F_IT_, or coefficient of inbreeding in relation to the total population; and F_ST_, or global inbreeding of the population) and their 95% confidence interval across loci. 

The Factorial Correspondence Analysis (FCA) was implemented in GENETIX v. 4.05 [28]. D_A_ genetic distances between the Balearic, Spanish, and international breeds were calculated with POPULATIONS v. 1.2.28 (Olivier Langella, Boston, MS, USA) [29], and a neighbour-net network of the populations was displayed in SPLITSTREE4 v. 4.14.4 (Huson, Tübingen, Germany) [30]. At the individual level, D_AS_ genetic distances were calculated using POPULATIONS v. 1.2.28 (Olivier Langella, Boston, MS, USA) [29], and a neighbour-joining individual tree was visualized with FIGTREE v. 1.4.4 (Rambaut, Edinburgh, Scotland) [31]. 

The population structures of Ca Rater Mallorquí and the Balearic, Spanish, and international breeds were investigated by the Bayesian model-based method of STRUCTURE v. 2.1 (Pritchard lab, Stanford, CA, USA) [32]. It was performed with an initial burn-in length of 200,000 followed by 500,000 MCMC iterations (Markov Chain Monte Carlo), with 10 repeats for K ranging from 2 to 19. The optimum K value was calculated according to the Mean Ln Probability method using Structure Harvester v. 0.6.94 (Dent A. Earl, Santa Cruz, CA, USA) [33]. The results were graphically represented by the software CLUMPAK v. 1.1 (Kopelman, Tel Aviv, Israel) [34].

## 3. Results

### 3.1. Genetic Diversity within Ca Rater Mallorquí Dog Population

If we consider that the PIC is an informative measure of microsatellite loci concerning He [35], our results (Appendix A) suggest that all markers were useful in evaluating the genetic diversity of Ca Rater Mallorquí dogs (PIC > 0.5).

All the markers were polymorphic, and the number of alleles per locus ranged from two at 0959RD and 1055RD to 15 alleles at REN169O18; Therefore, we can assume that the Ca Rater Mallorquí population has moderate allelic diversity. In fact, the average mean number of alleles was 6.61. Since the selection response is determined by the initial number of alleles [35], the effective number of alleles is relevant to the long-term evolutionary potential of the Ca Rater Mallorquí population. As expected, its mean was lower (3.58) than the mean number of alleles and should be considered a medium-high value. The genetic diversity of Ca Rater Mallorquí He (0.685) was slightly higher than the Ho (0.655).

Only four markers significantly deviated from the Hardy–Weinberg equilibrium after the Bonferroni correction (Appendix A). Although REN169D0, 0959RD, and 0914RD showed a significant excess of homozygosity, particularly marker 0669RD and its significant homozygosity defect, this did not translate into a significant heterozygosity deficiency for the entire population. Its F_IS_ value (F_IS_ = 0.044; confidence interval = −0.0002–0.073) revealed a controlled percentage of homozygosity within populations [36], suggesting that the Ca Rater Mallorquí did not significantly deviate from the Hardy−Weinberg equilibrium.

### 3.2. Genetic Diversity among Ca Rater Mallorquí and Different Balearic, Spanish and International Breeds

Genetically speaking, we must consider two reasons for the Ca Rater Mallorquí population’s breed differentiation. Firstly, its genetic equilibrium demonstrates no recent migrations, as discussed in the previous section. Secondly, analysing its genetic relationship with populations or breeds that may be commercially, geographically, or historically connected to Ca Rater Mallorquí helps establish its level of differentiation. This genetic relationship will be analysed in this section.

#### 3.2.1. Genetic Diversity among the Ca Rater Mallorquí and Different Balearic Breeds 

Wright’s fixation indices and their 95% confidence interval across loci after 1000 bootstraps among the Ca Rater Mallorquí and other Balearic dog populations (Table 1) were: F_IS_ = 0.021 (0.018–0.049), F_IT_ = 0.149 (0.157–0.191) and F_ST_ = 0.131 (0.130–0.160). The F_IS_ value reveals a controlled percentage of homozygosity within populations as a result of random mating. The F_ST_ value indicated a moderate differentiation among the Terrier populations under study [36,37], signalling that approximately 13% of the total genetic variation could be explained by between-breed differences, and the remaining 87% by differences between individuals. 

We used FCA to investigate the genetic differentiation between individuals in each Balearic population. The sum of the first three axes explained 72.03% of the total genetic differentiation. Considering the Podenco Ibicenco and Ca de Conills breeds, which were difficult to divide in all the axes considered, the clear separation between individuals of the breeds Ca Rater, Ca Mé, Ca de Bou and Ca de Bestiar was remarkable (Appendix A).

#### 3.2.2. Genetic Diversity among Ca Rater Mallorquí and Spanish and International Breeds

Wright’s fixation indexes and their 95% confidence interval across loci after 1000 bootstraps among the Ca Rater Mallorquí, Spanish, and international breeds were the following: F_IS_ = 0.034 (0.020–0.048), F_IT_ = 0.203 (0.185–0.212), and F_ST_ = 0.175 (0.161–0.190). The F_IS_ value represents a controlled percentage of homozygosity within populations as a result of random mating. The F_IT_ value indicated a relevant heterozygosity deficiency caused by the absence of gene flow among the 18 breeds under study. The F_ST_ value revealed an important differentiation among the populations [35,36], where 17.5% of the total genetic variation was due to differences between the breeds, and the remaining 82.5% was due to differences among individuals. 

In Table 2, the genetic diversity parameters of all the breeds are shown. Ca Rater Mallorquí displayed the second highest mean number of alleles. Podenco Valenciano was the breed with the highest allelic diversity and number of alleles. The allelic richness of Ca Rater Mallorquí was intermediate between the lowest value of Bull Terrier and the highest value of Podenco Valenciano. According to heterozygosity, Ca Rater Mallorquí presented a high value of He and Ho, similar to those found in the Podencos group. Terriers showed the lowest genetic diversity values, especially Bull Terrier. Only Ca de Bestiar, Ca de Conills, Podenco Valenciano, and Jack Russell Terrier had a significant excess of homozygotes (Table 2). The rest of the breeds, including Ca Rater Mallorquí, appear to be in the Hardy–Weinberg equilibrium.

The sum of the three FCA axes explains 37.26% of the total genetic differentiation (Figure 1). In axis 1, two groups are evident, with a genetic differentiation of 16.14%. One group integrated the Kurt Russell Terrier, and the other group integrated the rest of the breeds, with a slight differentiation in Ca Mé Mallorquí. Axis 2 reveals that Bull Terrier, Fox Terrier, West Highland White Terrier, and Pointer breeds are separated from the rest, whereas the Podenco Ibicenco, Podenco Valenciano, Podenco Andaluz, Podenco Canario, and Braco breeds were closest to one other. Ca Rater Maloquí is nearer to this group of breeds than to the “terriers”. The separation between German Shepherd, Ca de Conills, and West Highland White Terrier from the rest is shown on axis 3. The Ca Rater Mallorquí is close to the Podencos group and Jack Russell Terrier, as shown in Figure 1. Its proximity to some of the terriers may be explained by the British influence on the Balearic Islands, which favoured the crossbreed, especially with Jack Russell Terriers [37].

Complementing the FCA analysis is the neighbour-net representing D_A_ genetic distances between populations, which reported a division between the 18 breeds into two big clusters. The first one comprised German Shepherds, Ca de Bou, Ca de Bestiar, and all the terriers except West Highland White Terriers. The second group integrated the rest of the breeds, i.e., the Podencos group and other hunting breeds. Ca Rater Malloquí was in this second cluster in an intermediate position, although closer to the Podencos than to the grouping formed by Ca Mé, Pointer, Braco, and the West Highland White Terrier (Figure 2).

We used a Bayesian clustering approach to investigate breed structure and relationships with autosomal microsatellite data assuming that the observed genetic diversity results from the genetic contributions of a variable number of ancestral populations (K). The results of our analysis can be observed in Figure 3, where graphics of results between K2 and K18 are also represented. When K = 2, one cluster was formed by Ca Bou, Bull Terriers, and Kerry Blue Terriers, and another cluster integrated the rest of the breeds, including Fox Terriers and Jack Russell Terriers, a mixture of both clusters. When the number of ancestral populations was assessed at K = 3, a new cluster integrating Ca Rater Mallorquí, Ca Mè Mallorquí, West Highland White Terrier, and Braco breeds appeared, showing some Podencos and German Shepherds with different admixture degrees than the original cluster. The same distribution was observed in K4. The optimum K value was K11, when Ca Rater Mallorquí appeared in a differentiated cluster until the end of the analysis (K18). Podencos represented a homogeneous group, except for Ibicenco, which was rather different and shared some similarities with Ca de Conills. The Terriers did not cluster together, revealing a high genetic differentiation. However, it is worth noting that distinguishing between Fox and Jack Russell Terriers was impossible in this analysis. The Ca Rater Mallorquí had a homogeneous population with slight admixture in some isolated individuals and a small group of animals different from the rest of the breed at K18. Similar results were found in the individual genetic distances of the neighbour-joining tree, where most of the Ca Rater Mallorquí were grouped into the same cluster. Some individuals, however, had an unclear grouping (Appendix A).

## 4. Discussion

### 4.1. Moderate Level of Genetic Diversity within Ca Rater Mallorquí

Ca Rater Mallorquí is a rat-hunting dog breed whose origins can be exclusively traced to the island of Mallorca in the Balearic archipelago. The morphological characteristics of this breed have been linked to two ancient dog breeds: dogs with pharaonic origin and dogs with the terrier suffix [6]. Until now, they have not been explored in genetic studies. Ca Rater Mallorquí is strongly related to the Valencian rat hunting dog due to the arrival of rice growers from Valencia to Mallorca and the repopulation of Valencian villages such as Taberna during the seventeenth century with Balearic citizens who brought their dogs [5,38]. Due to the historic British presence in the Balearic Islands, it is believed that Ca Rater Mallorquí was influenced by the British Terrier, although this has not been confirmed. Our study includes some Terrier breeds to clarify their genetic relationship with Ca Rater Mallorquí. Additionally, we established its genetic differentiation from five other Balearic and international hunting breeds using German Shepherds as the outgroup. Therefore, the breeds we considered in our research were chosen for geographic coexistence and evolutionary theories. 

The Ca Rater Mallorqui was officially recognised by Royal Decree 558/2001 on 25 May 2001 [39]. The breed standard was published in the Official Journal of the Balearic Islands on 28 December 2002. The management of the breed registry was entrusted to the Club by Resolution on 21 November 2002. On 10 March 2004, the Ministry of Agriculture, Fisheries and Food officially recognised the breed, and the standard was published in the BOE [40]. 

More recently, the breed’s use as a pet has increased. This practice has partially changed the dog’s nature as a hunter and decreased the genetic diversity of the breed, similar to many local dog breeds globally.

Currently, the Ca Rater Mallorquí population is at risk of extinction. The absence of any conservation program endangers the survival of a breed rooted in rural life on the island of Mallorca. There are no previous studies on their genetic characterisation using microsatellite markers or other tools. Therefore, we aimed to provide data on diversity parameters, genetic structure, and genetic relationships between Ca Rater Mallorquí and other local breeds of the Balearic Islands that share similar uses or geographic location, as well as some Terrier and international breeds such as German Shepherd, Braco, or Pointer as evolutive outgroups of the breed’s formation and evolution. Our results will be the starting point to knowing the genetic situation of the breed and designing a conservation plan that offers preservation tools for the breed’s purity and increasing the population size.

Our findings showed that Ca Rater Mallorquí presents a moderate allelic diversity with a mean number of alleles (6.61) lower than was reported in the Podenco Valencian rat-hunting dog (7.9) [41] but similar to those found in the Spanish Water Dog (6.0) [42] and higher than in its island neighbour Ca de Bou (5.05). Since the selection response is determined by the initial number of alleles [43], this parameter is relevant to the long-term evolutionary potential of the Ca Rater Mallorquí population. As expected, its mean was lower (3.58) than the mean number of alleles per locus and should be considered a medium-high value between Pointers dogs (4.3) [44] and dogs with lower values, such as the English Bulldog (2.7) [45]. The allelic richness of Ca Rater Mallorquí was lower than the values found in some Italian Shepherd dogs [9] but similar to the Czechoslovakian Wolfdogs reported by Smetanová et al. [46]. 

The genetic diversity of Ca Rater Mallorquí (He = 0.685) is slightly higher than the Ho (0.655). These values represent a moderate genetic diversity and are lower than those reported in the Podenco Valenciano (He = 0.791; Ho = 0.714) [30] and African village dogs (He = 0.690; Ho = 0.770) [3], but higher than the Golden Retriever (He = 0.657; Ho = 0.567) [47]. Despite being an endangered breed reared on an island, Ca Rater Mallorquí does not significantly deviate from the Hardy–Weinberg equilibrium, nor does it show a significant excess of homozygosity. They have moderate to high levels of genetic diversity considering both allelic variation and heterozygosity, which may reflect efficient management of local populations counteracting inbreeding and genetic drift. Therefore, even if the breed was declared to be in danger of extinction, their present levels of genetic diversity are optimistic for conservation. 

### 4.2. Population Genetic Structure of Ca Rater Mallorquí

On the other hand, Ca Rater Mallorquí has not received recent influences from breeds with the same geographic distribution such as Ca Mè Mallorquí or Ca Bou, all from the Mallorca Islands, nor does it share the same functionality as Ca Mè Mallorquí or Ca de Conills de Menorca. These findings are supported by the FCA’s analysis of the six Balearic Island breeds and their relatively high values of F_ST_ (0.131). This population divergence is justified by the microsatellite markers used in this study because they were neither completely neutral nor linked to genomic regions under selection, as reported by Luikart et al. [48]. Other authors also found them in the canine species [41,42]. These are positive findings because traditional breeders suspected genetic erosion due to its recent use as a pet.

Compared with the rest of the breeds included in our research, a higher F_ST_ value was obtained (0.175), as expected. These values revealed an elevated differentiation among the populations [35,49], where 17.5% of the total genetic variation was due to differences between breeds, and the remaining 82.5% resulted from differences among individuals. These results could be the result of different origins and evolutions of the diverse groups of studied breeds. Although Terriers, Podencos, and other hunting dogs share the same functions, they appear in different clusters of the neighbor-net graphic of the D_A_ genetic distances and STRUCTURE analysis, as expected for their assignment to different dog branches (Graioids and Lupoids). In the same way, the FCA results separate them into two groups. One group integrated the Jack Russell Terrier, and the other group integrated the rest of the breeds with a slightly differentiation of Ca Mé Mallorquí. Axis 2 revealed that Bull Terrier, Fox Terrier, West Highland White Terrier and Pointer breeds were clearly separated from the rest, whereas Podenco Ibicenco, Podenco Andaluz, Podenco Canario, and Braco breeds were the closest to each other. Ca Rater Mallorquí is close to this group of breeds, the Podencos group, and Jack Russell Terrier. Their proximity to some of the terriers could be explained by the British influence on the Balearic Islands that favoured the crossbreed, especially with Jack Russell Terriers [37]. However, additional analysis is needed to confirm this statement.

According to our findings, the Ca Rater Mallorquí is a homogeneous population with no signs of recent genetic introgression. There is no evidence of substructure in the population, and we cannot assume that it has received some influence from Terriers as some studies in the literature suggest, probably due to the breeders’ intuitive genetic management in diluting this influence. This methodology for recovering anonymous animals is interesting since we are working with a low census and endangered breed.

Although microsatellite markers are useful in inferring similar biological processes and patterns to SNP, these results should still be confirmed using genome-wide SNP commercial arrays. Microsatellites and SNP offer similar results in population genetic studies, especially in populations with low genetic diversity [50].

## 5. Conclusions

In this study, we confirmed that Balearic breeds have remained pure and without crossbreeding even though two breeds, Ca de Conills and Podenco Ibicenco, show a closer genetic relationship because they share the same distribution area. The genetic diversity analysis of Ca Rater Mallorquí has provided us with their genetic profile as a homogeneous breed, according to the Hardy–Weinberg equilibrium, with moderate genetic diversity and without internal structure or varieties. They are totally differentiated from the Balearic, Spanish, and international breeds we studied. Their morphology and abilities as hunting dogs specialised in hunting small mammals have been the main reasons to associate the Ca Rater Mallorquí with Terriers. However, our study showed that they are not genetically related.

According to the Ca Rater Mallorquí census data, a conservation program should be designed and implemented to help breeders manage controlled and purebred crossbreeding. Our study results are already being used to manage the genealogy of the breed, verify genealogic registers, and ensure the genetic assignment of registered animals to the breed. There are many local breeds worldwide with the same endangerment situation as the Ca Rater Mallorquí. Therefore, our study can serve as a model to characterise, conserve, and evaluate this breed, which has served humanity for centuries, and now, in the present, with their role in sustainability.

## Figures and Tables

**Figure 1 animals-12-02733-f001:**
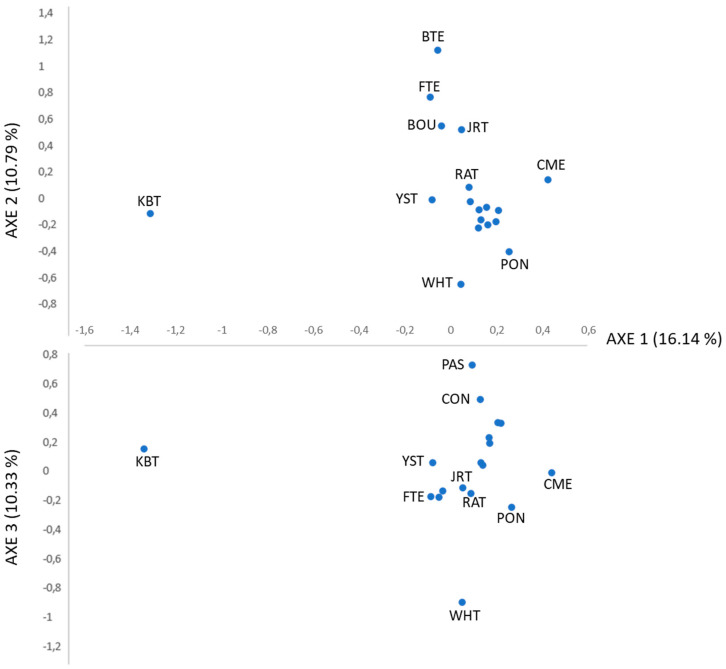
Spatial representation of genetic distances among 18 dog breeds from the first three axes obtained in the factorial analyses of correspondence based on microsatellite data. Values between brackets on axes represent the contribution in percent to each axis to total inertia. Breed acronyms are described below the Acronym title in Table 1.

**Figure 2 animals-12-02733-f002:**
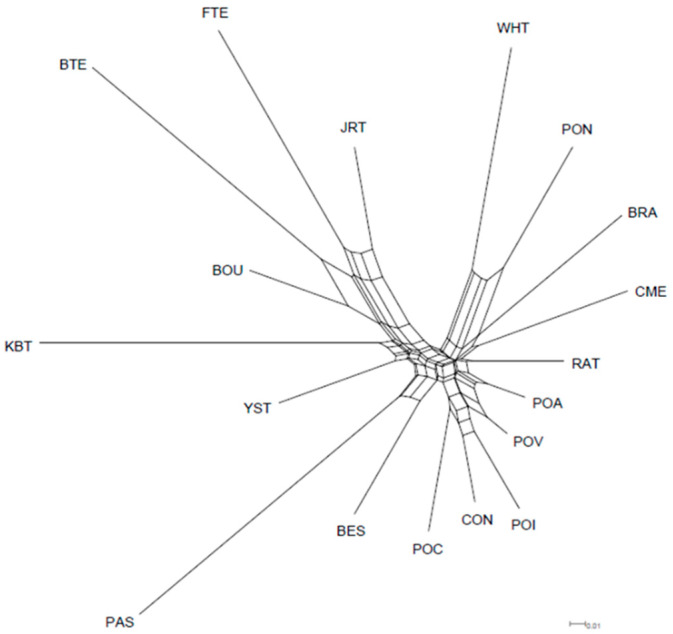
Neighbour-net representation of Nei’s DA genetic distances between 18 dog breeds based on microsatellite data. Breed acronyms described below the Acronym title in Table 1.

**Figure 3 animals-12-02733-f003:**
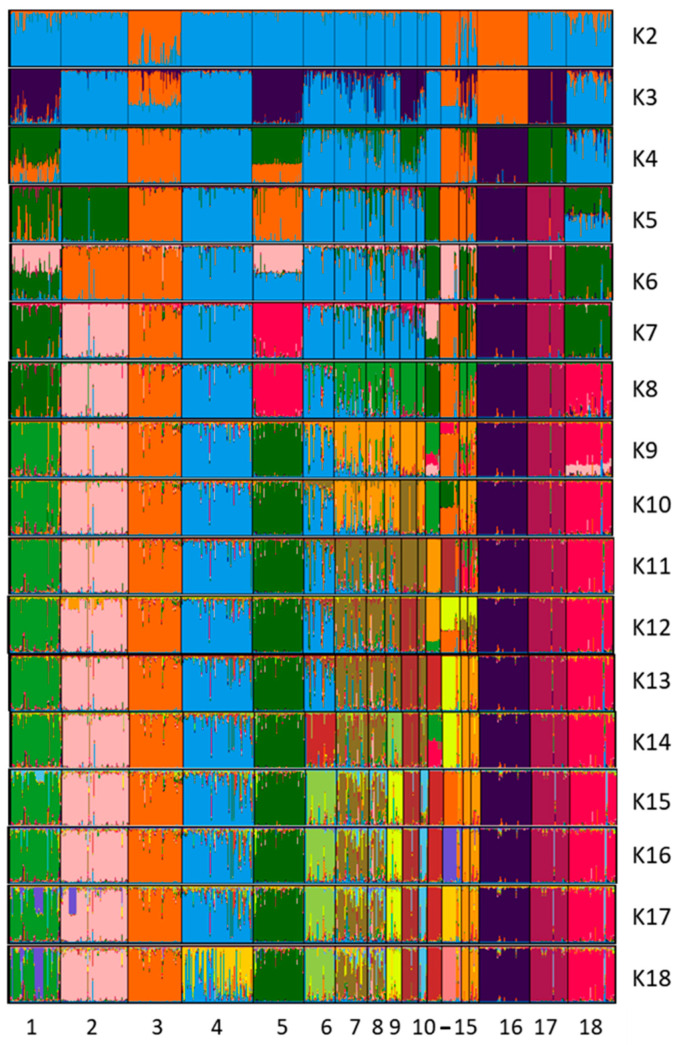
Population structure of 18 dog populations based on microsatellite loci using STRUCTURE software. Each breed is represented by a single vertical LINE divided into K colours, where K is the number of assumed ancestral clusters, which is graphically represented as K = 2 to K18. Breed acronyms are described below the Acronym title in Table 1.

**Table 1 animals-12-02733-t001:** Breeds included in the between-breed analysis. N: number of samples, FCI and RSCE represent classifications according the Fédération Cynologique Internationale (FCI) and the Spanish Royal Canine Society (RSCE), respectively. NR: Not recognised, * Balearic breeds, ** Podenco: Hound.

Breed (Acronym) *, **	N	FCI	RSCE
1 * CA RATER (RAT)	79	NR	NR
2 * CA DE BESTIAR (BES)	106	1	1
3 * CA BOU (BOU)	83	2	2
4 * CA DE CONILLS DE MENORCA (CON)	112	NR	NR
5 * CA MÈ MALLORQUÍ (CME)	80	NR	NR
6 *^,^ ** PODENCO IBICENCO (POI)	49	5	5
7 ** PODENCO VALENCIANO (POV)	50	NR	5
8 ** PODENCO ANDALUZ (POA)	29	NR	11
9 ** PODENCO CANARIO (POC)	25	5	5
10 BRACO (BRA)	26	7	7
11 GERMAN SHEPHERD (PAS)	14	1	1
12 POINTER INGLÉS (PON)	23	7	7
13 BULL TERRIER (BTE)	30	3	3
14 FOX TERRIER (FTE)	13	3	3
15 JACK RUSELL TERRIER (JRT)	14	3	3
16 KERRY BLUE TERRIER (KBT)	80	3	3
17 WEST HIGHLAND WHITE TERRIER (WHT)	60	3	3
18 YORKSHIRE TERRIER (YTE)	71	3	3

**Table 2 animals-12-02733-t002:** Genetic diversity parameters in 18 dog breeds from 20 microsatellites. Breed, number of samples (N), mean number of alleles (MNA), effective number of alleles (Ae), Allelic richness (Rt), expected heterozygosity (He), observed heterozygosity (Ho), F_IS_ and its 95% confidence interval across loci (CI). Breed acronyms are described below the Acronym title in Table 1.

Breed	N	NMA	Ae	Rt	He	Ho	F_IS_	CI
RAT	79	7.25	4.85	3.72	0.713	0.700	0.00953	(−0.02321–0.03284)
BES	106	6.95	4.67	3.88	0.712	0.706	0.12109	(0.07375–0.15422)
BOU	83	6.35	4.05	3.19	0.644	0.566	0.00804	(−0.13240–0.04131)
CON	112	7.20	4.7	3.73	0.712	0.686	0.26097	(0.09149–0.34834)
CME	80	6.70	4.23	3.27	0.675	0.675	−0.00095	(−0.03597–0.01973)
POI	49	5.75	4.26	3.44	0.699	0.718	0.19955	(−0.01783–0.25776)
POV	50	7.65	5.38	4.45	0.759	0.717	0.03798	(0.00756–0.05897)
POA	29	7.10	5.34	4.24	0.743	0.710	0.08302	(−0.05333–0.11452)
POC	25	5.90	4.6	3.46	0.674	0.690	−0.02857	(−0.08105–0.00964)
PON	26	5.30	4.17	2.96	0.612	0.614	−0.02982	(−0.11950–0.00741)
BRA	14	5.55	4.89	3.59	0.728	0.723	0.04539	(−0.02694–0.08134)
PAS	23	4.05	3.43	2.59	0.589	0.607	−0.02354	(−0.10593–0.01139)
BTE	30	3.70	2.82	1.89	0.428	0.318	−0.02687	(−0.08048–0.00434)
FTE	13	4.15	3.82	2.74	0.600	0.485	−0.00416	(−0.07732–0.02652)
JRT	14	5.05	4.48	3.3	0.665	0.612	0.05560	(0.01131–0.07960)
KBT	80	3.80	2.95	2.09	0.490	0.504	0.01938	(−0.02246–0.04994)
WHT	60	4.70	2.94	2.14	0.465	0.431	0.07309	(−0.00767–0.13856)
YST	71	6.90	4.71	3.77	0.713	0.688	0.03520	(−0.00873–0.06431)

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
