# Peer review of "Genetic Structure of the Ca Rater Mallorquí Dog Breed Inferred by Microsatellite Markers"

_animals, 2022, doi:10.3390/ani12202733_

Round 1

Reviewer 1 Report

The Ca Rater Mallorquí dog breed's genetic characterization was the main goal of the study in order to determine the breed's current status and the best strategy to conserve it. The contribution of this work is in the genetic profile of the Ca Rater Mallorquí breed described for the first time. The methodology is appropriately chosen with sufficient number of tested samples.

Approximately 25 % of citations are published recently within five years. And around 59 % are from the last fifteen years. Considering the topic, I think that it is a sufficient amount of recent references.

Although I do not feel qualified enough to judge about the English language, I still think minor revision is needed due to errors in English. Below are some of my observations.

The second preposition “of” in the title should be removed, see line 2 (Genetic structure of the Ca Rater Mallorquí breed). It is better to unify written English. In my opinion, write a preposition “the” before the breed name, see line 22 or 113 and use either “heterozygosity” (see line 246) with “heterozygosities” or “heterocigocity” with “heterocigosities” (see line 255) throughout the text.

Occasionally incorrect words are used in the text. An incorrect word “witch” on the line 71 and 352 should be changed to “with”. Furthermore, the word “it”, see line 81 and 106, should be changed to “its”. The word “demands”, see line 65, should be changed to “demand”. The word “outcoming”, see line 230, should be changed to “outcome”. The word “that to”, see line 295, should be changed to “than to”. The word “no possible”, see line 320, should be changed to “not possible”.  The word “pray”, see line 347, should be changed to “prey”. The word “probed”, see line 349, should be changed to “proved”. The word “stablished”, see line 359, should be changed to “established”. The preposition “of” before the dates, see line 365, 367 and 368, should be changed to “on”. The word “breed”, see line 371, should be changed to “breed’s”. The word “blanches”, see line 382, should be changed to “branches”. The word “huntingo”, see line 389, should be most likely changed to “hunting dog”. The word “this local populations”, see line 406, should be changed to “these local populations”. On line number 417 and 418, the preposition “a Luikart” should be deleted and replaced with “by Luikart”. The word “results separates”, see line 428-429, should be changed to “results separate”. The word “close the Podencos group”, see line 434, should be changed to “close to the Podencos group”.

There is an error in the Figure 3 title, see line 326, “pig” should be changed to “dog”. There is a typing error in the first supplementary table named “Supplementary table 1”, “Crhomosome” should be changed to “Chromosome”. All abbreviations used should be mentioned in the description of the tables. For instance, in the Table 1 and Table 2 there is no explanation of the abbreviation “N”. The description of the Table 2 does not clearly state that “Fis IC” used in the table is for the “confidence interval”. The abbreviation “KRT” used in Figure 1 is not explained as well. I suppose it is an abbreviation for the breed “Kurt Russell Terrier”.

The sentence should end with a dot on line 467. In the references, there are two dots at the end of lines 500 and 571. The years of publication on the line 540 and 556 should be in bold, as they are for other citations.

Overall, I think the authors did a great job and only minor proofreading is needed.

Reviewer 2 Report

This paper delves into the genetic structure of a local dog breed (Ca Rater Mallorquì) using microsatellite markers.

It is not unlike many other published papers in which similar examinations are undertaken, albeit for different breeds. Genetic characterization of animal genetic resources is a major issue in animal breeding. Therefore, the study may have interest.

In general, it is well written but some issues need attention before it can be accepted for publication.

However some comments are due. The authors justify the use of microsatellites (Lines 93-101), but these markers became obsolete for these studies. Commercial SNP chip are available for dog. More than 150,000 loci (170K genome-wide SNP array) give lots of information about the genome and they are free from genotyping errors like microsatellites. This aspect should be reported in the discussion.

Moreover, for comparison purpose, genotype data from other dog breeds are available from important project, as LUPA (http://dogs.genouest.org/SWEEP.dir/Supplemental.html).

An improvement of the English spelling is needed along the whole manuscript. There are phrases with spelling and grammatical errors (e.g. Line 255, 259, 406, ….).

The statement contained in L439-441 needs additional analyses to be confirmed.

I found odd the total absence of analyses at the individual level; this kind of analyses can give complementary information supporting the main conclusions of a report.

A neighboring joining tree among individuals based on proportion of shared alleles could be added to the manuscript.

The distribution of the Ca Rater Mallorquì genotypes (at individual level) on a bi- or tri-dimensional space with the other breeds can illustrate if the differentiation reported is "real" or not.

The significance of the fixation indices can tested through the locus-by-locus analysis of molecular variance (AMOVA) procedure among breeds.

 The Discussion section can be shortened too (Lines 344-374).

The work does not provide enough valuable recommendations for the breed management and conservation. 

 Lines 412 an3 and 428: FCA?

I encourage the authors to revise and resubmit the manuscript, taking the reviewers' comments into consideration.

Reviewer 3 Report

The manuscript reported a simple genetic study on the Ca Rater Mallorquí dog breed based on microsatellite markers. The results provide information on the genetic background of this local dog breed. And thus, the manuscript deserves publication. However, several issues should be addressed during manuscript revision.

 Major issues:

1. The English expression of the manuscript should be significantly improved throughout the text.

2. The “Discussion” section should be better reorganized using two subtitles. I would like to suggest the authors use “4.1 Moderate level of genetic diversity within Ca Rater Mallorquí” and “4.2 Population genetic structure of Ca Rater Mallorquí”.

3. Considering the hair samples with Chelex DNA extraction, it is better to check whether there was an allele dropout for each microsatellite marker.

4. The “Conclusion” section should be shortened to one paragraph and should summarize the key findings, limitations, and future perspectives, not include the general description.

Minor issues:

5. The manuscript title should be corrected to “Genetic structure of the Ca Rater Mallorquí dog breed inferred with microsatellites markers”.

6. In simple summary, the sentence “The main aim of this study was the genetic within and between breeds characterization of Ca Rater Mallorquí dog breed to establish its present status and the way to preserve the breed” should be rewritten.

7. In lines 119-123, the breed names should be deleted, but mention 17 breeds included in Table 1. Other necessary information related to all breeds analyzed should be included in Table 1, not in the text.

8. The legend of Figure 3 mentioned “Population structure of 18 pig population”, this study worked on dog breeds, NOT pig populations. In addition, the population codes can be put in Table 1.

9. The authors should be more cautious when interpreting the results about whether the Ca Rater Mallorquí is a pure and well-defined breed because the power of around 30 microsatellites is limited and genome-wide SNP data is needed to further confirm the results.

Round 2

Reviewer 2 Report

The authors have significantly improved the text and satisfactorily responded to most of points raised by the reviewers and I appreciate your careful consideration of the comments provided.

Thus, I do recommend this manuscript for acceptance in Animals.

Reviewer 3 Report

The authors addressed all my concerns and thus I would like to recommend to accept it for publication.